# Mindfulness and Other Simple Neuroscience-Based Proposals to Promote the Learning Performance and Mental Health of Students during the COVID-19 Pandemic

**DOI:** 10.3390/brainsci11050552

**Published:** 2021-04-27

**Authors:** Gonzalo R. Tortella, Amedea B. Seabra, Jorge Padrão, Rodrigo Díaz-San Juan

**Affiliations:** 1Center of Excellence (CIBAMA-BIOREN), Universidad de La Frontera, Temuco 4811230, Chile; 2PhD Program in Natural Resource Sciences, Universidad de La Frontera, Temuco 4811230, Chile; 3Center for Natural and Human Sciences (CCNH), Federal University of ABC (UFABC), Av. dos Estados, 5001-Bangú, Santo André 09210-580, SP, Brazil; amedea.seabra@ufabc.edu.br; 4Centre for Textile Science and Technology (2C2T), University of Minho, 4800-058 Guimarães, Portugal; padraoj@2c2t.uminho.pt; 5Educational Neurosciences and Psychology Department, Centenario School, Temuco 4810936, Chile; rdiaz@colegiocentenariotemuco.cl

**Keywords:** COVID-19, mental health, stress, learning, education, neuroscience

## Abstract

The COVID-19 pandemic has had a negative impact on education. The restrictions imposed have undoubtedly led to impairment of the psychological well-being of both teachers and students, and of the way they experience interpersonal relationships. As reported previously in the literature, adverse effects such as loneliness, anxiety, and stress have resulted in a decrease in the cognitive performance of school and higher education students. Therefore, the objective of this work is to present a general overview of the reported adverse effects of the COVID-19 pandemic which may potentially influence the learning performance of students. Some neuroscientific findings related to memory and cognition, such as neuroplasticity and long-term potentiation, are also shown. We also discuss the positive effects of the practice of mindfulness, as well as other simple recommendations based on neuroscientific findings such as restful sleep, physical activity, and nutrition, which can act on memory and cognition. Finally, we propose some practical recommendations on how to achieve more effective student learning in the context of the pandemic. The aim of this review is to provide some assistance in this changing and uncertain situation in which we all find ourselves, and we hope that some of the information could serve as a starting point for hypotheses to be tested in educational research and their association with neuroscience.

## 1. Introduction

For everyone today, 19 December 2020 will be an unforgettable date. Severe acute respiratory syndrome coronavirus 2 (SARS-CoV-2), which causes the respiratory disease COVID-19 and is a highly transmissible pathogenic viral infection, was detected in Wuhan, China, whence it quickly spread around the world [1]. From the beginning, governments worldwide imposed movement restrictions and other policies requiring varying degrees of confinement as protective measures to stop or slow down transmission of COVID-19. Our daily lives changed abruptly, affecting our mental health and well-being. Active workers were obliged to remain physically absent from their worksites. In “teleworking” conditions, they were more susceptible to changes in mental health, specifically psychological distress and depression [2,3]. In the context of COVID-19, threats to mental health have received increased importance. Psychological impacts have been associated with adverse mental health consequences such as stress, depression, frustration, anxiety, and insomnia [4,5,6,7,8]. Post-traumatic stress symptoms associated with the pandemic have also included negative changes in cognition or mood, as well as hyperarousal [9]. Moreover, fear of the contagion has been highlighted as a key factor in emotional disorders and in the genesis of psychopathologies [10]. Although, to date, few studies have been carried out, the prevalence of generalized anxiety disorder (GAD), depression, and poor sleep quality have been reported as 35.1%, 20.1%, and 18.2% respectively. GAD and depression have been found to be significantly higher in younger people, associated with time spent worrying about SARS-CoV-2 (≥3 h/day) [9]. In this same context, it has been reported that healthcare actions during the COVID-19 pandemic are needed by patients with physical needs, and also for monitoring their psychological needs by providing supportive strategies for the short and long term [11].

A significant further concern is raised by the changes adopted in education, in both schools and higher education institutions. From April 2020, more than one billion learners were affected by the disruption caused by the pandemic [12], when schools were closed as a prevention policy. The real impacts on the mental health and learning performance of these students are still unknown; however, it has been suggested that their effects could continue into the future, even after the pandemic is brought under control [13]. The psychological impact has been investigated by measuring the anxiety level among a sample of 7143 college students, in which approximately 25% of the students showed some degree of anxiety [14]. Although isolation periods have varied by country, they have meant that in-person teaching had to be replaced by virtual teaching alternatives to try to continue normal academic activities [15]. There is no doubt that in a digital age, technological media is an emergent tool that can be used to enhance both teaching and learning capacities, however, the interruption of normal classroom activities is a new and complex scenario for both teachers and students [16]. Another important aspect in this complex scenario is that educational plans are designed mainly for in-person teaching, not online. It has been reported that there is an urgent need for teachers to learn how to use emerging technology more effectively, to make them more efficient instructors [17]. Other aspects must also be considered, particularly socioeconomic differences between students which directly affect their ability to acquire technological tools for e-learning. In an analysis of distance education during this pandemic, the principal criticisms were directed at slide-based presentations, classical instruction, teacher-oriented question-and-answer, stationary content-based instruction, and the inability to evaluate and monitor [18]. According to [15], the main idea is to focus the process on student assessment considering complete learning (i.e., use of concept mapping, problem-based learning discussions, class presentations, peer-teaching, etc.). Getting the students involved and establishing learning challenges for them that mean more than just a final grade, allows them to take risks without the fear that their failure will be punished (bad grade); thus, they gain resilience and are motivated and enthusiastic about a new attempt, increasing their learning ability. This has previously been recommended in an educational context by [19] in their book “*Neuroscience and Education*”. Undoubtedly, as [15] noted, this assessment involves hard work for the teacher, especially in the context of online learning; however, knowledge of how each student learns is necessary for a more efficient process, and neuroscience can offer guidance in this important aspect [20,21]. As [20] indicated, the relationship between neuroscience and classroom practice has been viewed with some scepticism; however, with current advances, the value of neuroscience in learning is becoming more widely accepted. For example, it has been scientifically proven that memorizing or repetition in the classroom does not facilitate learning [22]; more promising results are achieved by doing and experimenting, and therefore generating excitement [23]. Other important aspects considered by neuroscience, such as nutrition, physical activity, and good sleep habits [19,24], which are also directly related with learning, must also be taken in account, as discussed below. As mentioned by [25], teaching strategies should be applied that encourage students to use brain processes that support the formation and retrieval of long-term memories, which is decisive in the effectiveness of teaching and learning. In this context, we must understand long-term memory as our capacity to store knowledge (information) for a long period of time, referring to all the information that can be retrieved consciously (explicit memory) or unconsciously (implicit memory) as indicated by [26]. The term “memory consolidation” is closely related to the previous term, referring to the necessary process by which a labile or temporary memory can be transformed into long-term memory [27].

In this work, therefore, we describe how both our mental health and our learning capacity have been affected by the effects of the COVID-19 pandemic. We also address several factors that have been proposed as learning stimulators by neuroscientific studies, such as restful sleep, good nutrition, physical activity, and the practice of mindfulness. In view of this, we address how to promote mental health resilience by applying these protective factors, thus, favouring learning capacities. Finally, we present some guidelines on how to transmit knowledge to students in the scenario of the current pandemic, taking into account the protective factors mentioned above. It should be noted that all the physiological or molecular processes in our brain involved in learning and memory retention are very complex and have been reviewed extensively [28,29]; it is not our intention to discuss this enormous field in detail again here. Moreover, the considerations, reflections, and guidelines presented in this work are not an absolute solution to the learning problems associated with the mental health challenges facing students in the pandemic; rather, it is our intention to contribute some possible interventions and strategies drawn from neuroscience that could be applied, or that could serve as a basis for studies to explore alterations in students cognitive capacities resulting from the pandemic. They may also assist teachers and parents seeking how best to support students during and after the COVID-19 pandemic, or in possible future stress situations.

## 2. COVID-19 Outbreak and Adverse Effects That Potentially Influence the Learning Performance of Students

The COVID-19 viral disease has changed our lives through the many restrictions imposed by governments to reduce contagion, such as lockdown, social distancing, and the closure of schools and higher education institutions. As mentioned above, more than one billion students have had to transfer their studies to their homes; likewise, teachers have had to teach their classes from their homes, online, without the advantage of prior preparation or guidelines on this mode of teaching. This process has not been easy for anyone. Therefore, we ask whether we are really prepared to teach or learn under e-learning conditions; and in the physical and emotional context of the outbreak, are students’ brains prepared to capture the information studied and retain it in the long term? To begin with, it must be understood that learning is the acquisition of new information, where both internal and external stimuli play an important role in the synaptic activity in our brains, and therefore also in the learning processes [30]. It should be noted that the term “learning” involves broader aspects than education. However, in this work, we use it in the pedagogical context. Other aspects that can also influence learning are age, ability, and prior knowledge [31]. The processes of memory creation and learning are complex and are based on dynamic interactions between distributed brain areas, as well as innumerable neuronal networks where additive network interactions can occur [32,33]. One factor is that challenging events in the environment can initiate a stress response consisting of a cascade of adaptive psychological and physiological processes [34,35], however, this area is outside the scope of the present work (interested readers can review [35] or [36]). Since the learning process in our education is easily altered, it is not surprising that emotions or sensations triggered by life events can affect mental health [37], and therefore learning performance, as shown in Figure 1. The author of [38] recommended several considerations directly related with our mental health wellness that should be taken into account during the COVID-19 outbreak. Stress, depression, anxiety, mood alteration, frustration, etc. are emotional states with an enormous capacity to influence processes closely related with memory and learning. In this regard, learning and memory under stress and their implications for the classroom have been extensively reviewed by [39]. Both retention and learning are closely related to synaptic efficiency (quality and intensity of stimuli), i.e., they have a high sensorial and emotional potential, as well as frequency, and therefore generate new neural activity patterns [40]. However, it has been shown that stress and other negative emotions can act as inhibitors, blocking cognitive processes [30]. In fact, it has been reported that the neuronal networks activated by stress are closely related to the neuronal networks associated with cognitive processes [30].

Given that human emotions comprise complex interactions and promote both reasoning and the retention of information in our brain during learning [41], the lack of emotional regulation, triggered by negative emotions caused by lockdown, loneliness, or social distancing is undoubtedly able to trigger a decrease in learning performance [42]. A study carried out in primary school pupils in Argentina revealed that pupils in lockdown presented emotional regulation problems that had a negative effect on their comprehension capacity [43]. Moreover, these authors reported that teachers have difficulty assisting their pupils with aspects that were mostly related to their emotions. A study carried out in United States demonstrated that increased stress, depressive thoughts, and anxiety due to the COVID-19 outbreak were detected in college students [44]. The authors reported that the main causes were fear and worry, related with the students’ own health or that of their loved ones, leading to difficulties with concentration, sleep alteration, and increased concern about academic performance. Undergraduate and postgraduate students from West Bengal, India, have likewise had to deal with COVID-related problems, such as depression, anxiety, an unfavourable study environment at home, and poor internet connectivity [45]. These authors suggested the need to create a positive work environment for study, as well as strategies to build a resilient education system in order to promote skills development in young minds. Teachers must play a fundamental role in learning management. Similar results in university students from France were reported by [46]; the authors reported that more than 50% of students were affected by stress or anxiety due to COVID-19 confinement strategies. Similarly, an evaluation of Bangladeshi university students showed that between 15% and 46% reported depression, anxiety, or stress [37,47]. For students, social distancing, understood as separation from their peers due to the temporary cessation of classroom activities, generates loneliness and a decrease in the positive emotions that can stimulate learning. Furthermore, the lack of opportunities for interacting in-person with their peers implies the absence of the stimulus that triggers learning based on competition or social influence [48], which are also important in developing an interest in specific courses [49]. A recent study reported that during the pandemic, social presence was the most important factor affecting learning processes in students at higher education institutions [50]. The study also revealed that students lost motivation and performed poorly using online learning methods, while the situation improved during in-person lessons. This agrees with [51], who reported that students easily lost motivation because they did not have their teachers and classmates around them for physical encouragement. However, it must be noted that as every individual is different, the effects of the pandemic on mental health are also individual. Contrary results have also been reported, for example, [52] reported a positive reaction of French students to confinement, with more commitment and similar results were reported by [53] in a study carried out in students at a high school in Madrid, Spain.

Every day we understand more about the important role of emotions in fostering effective learning. As mentioned above, it is evident that alterations in our normal life can trigger negative emotions that in the long term will affect our brains, and specifically, our affective neuronal networks, which are directly associated with triggering interest and motivation or, contrarily, anxiety and stress [54]. Affective neuronal networks are important in learning processes, given that they are interconnected with other major networks such as recognition and strategy networks [54], which together form part of learning.

The COVID-19 pandemic has presented us with an unprecedented public health crisis, which is undoubtedly affecting the mental health of the population, and students at schools or higher education institutions are particularly vulnerable to its effects. As mentioned above, the main symptoms are anxiety, stress, and depression, which clearly affect students’ capacity for concentration and effective learning. Therefore, an important task for teachers (in the educational environment) and parents (in the home) is to support students by providing tools that promote learning, and also inhibit the negative emotions that act as a threat, blocking cognitive processes. Educational techniques such as activation of prior knowledge, motivation, effective school practices [55], flipping the classroom [56], etc. have proven to be effective for learning. However, given that emotions play multiple important roles in an educational context, it is essential to know how to promote and maintain positive emotional states in students, the main goal being to favour optimal learning in a difficult context. Learning based on neuroscientific findings could be a complementary strategy to help students to deal with mental health problems and other difficulties that are detrimental to learning and have been imposed by the COVID-19 confinement period.

## 3. Learning Stimulation Based on Neuroscientific Findings

Learning is a complex process that for many years has been conceptualized from different points of view depending on whether the focus has been on sociological, psychological, or biological processes, or a combination of these [57,58,59,60]. This work does not aspire to validate or to criticize definitions or teaching techniques, but rather to show readers how they might harness the knowledge of neuroscience, combined with innovative teaching techniques, to promote learning processes and the creation of long-term memory, principally when students are subjected to complex psychological states such as those imposed by the pandemic.

Neuroscience, also known as neural science, is the study of the development, structure, and functions of the nervous system. Efforts are particularly focused on the brain and how it creates emotional responses (affective neuroscience) or cognitive responses (cognitive neuroscience), including all the biological and chemical processes that participate in the formation of these responses. However, the interdisciplinary nature of this field is also recognized in other branches [61]. In an educational context, the main objective is the creation of knowledge through the learning process; this is also interdisciplinary, requiring the incorporation of multiple scientific fields such as psychology, neuroscience, biology, education, and nutrition [62].

Neuroscientific findings have allowed us to learn how a series of events in the brain are associated with environmental factors as well as emotional and cognitive states, initiating internal responses which can favour or retard learning processes. According to [25] and [63], among others, learning in many species is linked to the central nervous system. It has been demonstrated that the hippocampal region, a complex structure embedded deep in the temporal lobe, plays a key role in cognitive processes and learning which, in turn, are associated with changes in both the strength and number of synaptic connections between neurons [64,65,66]. These are associated with the release of glutamate, dopamine, acetylcholine, and other neurotransmitters, which reinforce the formation of new synaptic connections and are also associated with motivation and attention tasks [25]. Other neurotransmitters such as cortisol, which may be released under stress situations, appear as blockers in learning and retrieval processes and play a role in memory impairment [67,68]. Learning is also associated with brain metabolism development [63,69,70] and the promotion of neurotrophic factors [71,72], which participate in the protection and repair of damaged neurons and the development of new neurons [63].

Another interesting neuroscientific finding closely related with learning and memory is neuroplasticity, which explains how the activity of the nervous system can change under varied stimuli (external or internal), leading to reorganization of its connections, functions, or structure [73]. The brain has the capacity to receive and process a continuous flow of stimulation from the environment, from which it is able to perceive stimuli, think, learn, remember, etc. Thus, experience can mould synaptic connections of the nervous system either functionally or structurally [74]. Neuroplasticity, also called synapse plasticity, involves the strengthening of some synaptic connections but also the elimination of others, as changes occur in the strength and number of connections between neurons when a significant stimulus is generated repeatedly [25,74,75], as shown in Figure 2. Synaptic plasticity, for example, by long-term potentiation remains the prevailing cellular model for learning and memory [76].

Neuroplasticity is a complex reorganization process which can occur at the tissue level up to molecular changes in the nervous system [77]. Enzymes such as protein kinases, protein phosphatases, and proteases play a vital role in brain plasticity at molecular levels, as well as brain-derived neurotrophic factors (BDNF) which play an important role in regulating synaptic plasticity [28]. Another molecule known for its participation in neuroplasticity processes is nitric oxide (NO), which has a key role in maintaining the processes associated with the opening of the NMDA receptor, the induction of dendritic spine growth, and the regulation of presynaptic plasticity in GABAergic and glutamatergic neurons [78]. Neurophysiological and molecular mechanisms that induce neuroplasticity involve complex processes, which continue to be studied today. This area is outside the scope of the present work and for more detailed information about the molecular mechanisms involved in neuroplasticity see [28,29]. In vitro and in vivo electrophysiological recording, as well as real-time monitoring of synaptic spine morphology changes by modern imaging technologies, have reinforced the idea of a link between synaptic change and memory storage, as mentioned by [79], who disclosed long-lasting and activity-dependent changes in synaptic efficacy. According to [79], long-lasting changes include long-term potentiation (LTP) (Figure 2) and long-term depression (LTD), which are expressed as changes in both pre-synaptic and post-synaptic elements. These are especially important in learning, because they are involved in brain development and recovery as well as long-term memory consolidation. In simple terms, when a student tries to memorize by reading without allowing time for reflection on or practice of the contents (solving problems for example), the long-term memory may not be consolidated. When something is studied and learned, synaptic connections occur between specific neurons. If what is learned is repeatedly put into practice, then, co-activation of these neurons also happens repeatedly, increasing the strength and number of connections. These new synapses can last for long periods of time, and the content learned is maintained in the long term (memory consolidation); it can easily be recovered by reactivation of the same connections between the same neurons, given that a neural substrate has arisen [25]. This process is particularly related to the ”forgetting curve” proposed by Ebbinghaus, which suggests that the people tend to forget newly learned knowledge in a short time unless they actively review the learned material [80]. This in turn is closely related to the theory of Heebs [81], which proposes that long term memory consolidation or LTP is achieved by repeated experiences. For more specialized reading see [25,73,75,82].

In addition, according to current neuroscientific knowledge, significant learning requires several factors which can act to block or counteract the expression of negative emotions (Figure 1) and promote learning, known as protective factors (Figure 3). It has been proposed that these protective factors (together or independently) play a fundamental role in stimulating learning because they can limit the deleterious effects of complex situations in life, such as anxiety, loneliness, or stress. As Figure 3 shows, the evidence indicates that physical activity, restful sleep, good nutrition, and mindfulness-based stress reduction can be effective tools for learning development [63,83,84,85,86,87,88,89,90,91,92]. In this context, a recent study suggested that the relationship between certain protective factors and mental health during the COVID-19 pandemic is complex and does not respond to a simple linear model [93]. Study of these protective factors as intervention tools aimed at improving mental health and learning could have important implications in the future.

### 3.1. Physical Activity and Cognitive Processes

Although studies on the relationship between physical activity and cognitive processes are still in its infancy, it has been reported that regular physical activity can trigger a reduction in depressive stages, increasing positive emotions and delaying cognitive impairment, and thus enhancing and maintaining brain functions [94,95,96]. In the hippocampus, physical activity has demonstrated an increase in hippocampal theta amplitude, as well as memory function [97]. Induction of neurotransmitters and BDNF has also been reported [96], which affects neuroplasticity by facilitating long-term potentiation (LTP) and dendrite formation in neural circuits, supporting learning [96]. Increased BDNF gene expression due to aerobic exercise can occur in multiple regions of the central nervous system, such as the hippocampus, cerebellum, spinal cord, and cerebral cortex [95]. An interesting recent work [29] reviewed the neurophysiological and molecular mechanisms of neuroplasticity that are induced by physical activity.

Physical activity is known to enhance memory and learning, however, this does not necessarily mean running or high-intensity activities. Exercise is correlated with psycho-physiological stress, increasing the arousal level and favouring the secretion of cortisol [98], which can act as a blocker in cognitive processes as mentioned above. It has been reported that simply pacing up and down the room while learning or analysing a problem is a great help [85,99]. Very low-intensity physical activity, such as walking, has been shown to improve recall in foreign-language vocabulary learning, as compared with encoding during physical rest [85]. Arithmetical operations were also enhanced in primary school pupils by physically active math lessons as compared with remaining seated in the classroom [92]. Some basic cognitive processes, i.e., perception, attention and memory, as well as executive functions develop in our brains mainly associated with mental processes that need attention and concentration, i.e., solving problems, creating plans, etc. [100,101]. Almost ten years ago, [102] reviewed, in detail, the effects of physical activity on executive functions; they found positive relationships between physical activity and executive functions, especially consistent with inhibitory control (deliberate suppression of impulsive responses), although other functions showed occasional or no effects. More recently, on the one hand, positive effects of physical activity on executive functions have been reported in preschool children (age 3–5 years), kindergarten children, children aged 3–7 years, preadolescent pupils (6–12 years), and older adults [2,103,104,105,106]. On the other hand, another work reported no significant effects of physical activity (Daily Mile™) on fluency in math or the executive function in children (mean age = 8.99 ± 0.5) [107].

### 3.2. Nutrition and Cognitive Processes

An emergent interdisciplinary field of research is nutritional cognitive neuroscience, which aims to evaluate the impact of nutrition on memory and cognition, as well as on brain health [108]. In the last few years, the study of nutritional factors has shown that they are important for neuronal function and synaptic plasticity; vital mechanisms have been revealed by which dietary factors trigger effects on mental function and brain health. Multiple brain processes such as regulation of neurotransmitter pathways, synaptic transmission, membrane fluidity, and signal-transduction pathways have been shown to be influenced by nutritional factors [83]. As these authors mentioned, some gut hormones or peptides such as leptin, ghrelin, glucagon-like peptide 1, or insulin, which act at the brain level or are produced in the brain, can influence cognitive processes. Several of these molecules can be identified in areas of the brain such as the hypothalamus, the cerebral cortex, and the hippocampus [83]. In addition, the same authors indicated that BDNF or synaptic plasticity regulators, which respond to peripheral signals such as food intake, can function as metabolic modulators. For more detailed information about nutritional cognitive neuroscience or the effects of nutrients on brain function, see the reviews [58,108].

It has been pointed out, many years ago, that good nutrition, in addition to physical activity, are important factors for the development of the cognitive capacity [83]. Foods have been shown to influence the plasticity of the central nervous system and its ability to recover from injury [109], as reported by [110]. These authors carried out an extensive review of how omega-3 fatty acids, polyphenols, caloric intake, and the combined effects of diet and physical activity can influence the cognitive function of our brains, increasing resistance and facilitating synaptic transmission in neurons; they are also very important for health, since a good diet favours the presence of molecules that play an active role in metabolism and in synaptic plasticity [110]. More recently, it has been reported that a diet enriched in polyphenols is a valuable influence on the central nervous system [111]. Findings from investigation into the effects of micronutrients on the brain have shown that selenium deficiency can cause oxidative stress and impaired cognitive functions [112], while calcium and iron play important roles in neuronal function [113]. A meta-analysis carried out by [114] evaluated the effects of supplementing diet with micronutrients on cognitive performance among school-aged children (4–18 years age) and reported that fluid intelligence was improved in micronutrient-deficient children. This has been consistently reported, but more evidence is needed for other cognitive domains, as well as for the effects in healthy children (without micronutrient deficiency). For more detailed studies of the evidence on the effects of micronutrients on brain development see the review offered by [115].

As mentioned above, negative emotional states such as stress or anxiety influence cognitive processes, which are also related to diet. Adequate consumption of fruit and vegetables has been demonstrated to result in more positive emotional states in Brazilian adults (lack of nervousness/stress) [116]. However, it is not easy to understand how our brain works and reacts to various situations, since it has also been reported that it is subjected to stressful situations due to complex psychological and physiological interactions caused by the desire for reward, leading to an increase in the consumption of unhealthy foods [117].

### 3.3. Restful Sleep and Cognitive Processes

The precise function of restful sleep or restful wakefulness has not yet been elucidated, however, both states are characterized by increased activation of brain structures, including the medial temporal and medial frontal regions associated with learning processes and memory [118,119]. By the same token, sleep deprivation significantly damages several cognitive and brain functions, particularly episodic memory and the underlying hippocampal function [120]. It has been suggested that recovery sleep lasting several hours can restore hippocampal synaptic plasticity and connectivity after brief sleep deprivation [121]. For more detailed information, [122] reviewed the impact of restful sleep on the efficacy of memory storage and on brain sections such as the hippocampus, striatum, amygdala, and thalamus, as well as the neuronal processes involved in these.

An association between restful sleep and learning consolidation has often been highlighted. Regrettably, sleep cannot be defined as a simple process where the cessation of physical activity allows us to rest, recover, and consolidate what has been learned from the tasks carried out previously. Perhaps this is the case in very simple terms, but sleep is really a dynamic and highly complex process involving genetic, biochemical, physiological, and psychological processes that allow memory consolidation [90,122,123,124]. For many years, a significant number of findings have proposed that sleep plays an important role in memory consolidation (see reviews carried out by [90,125]). Moreover, it has been reported that sleep has the additional function of restoring learning capabilities [126]. In addition, it has also been proposed that similar consolidation mechanisms can be achieved in some waking states, such as quiet, restful wakefulness [90]. A recent study evaluated whether restful sleep bears any relation to depressive states and a higher-grade point average (GPA) among college students (mean age 20.9 ± 1.5 years) [127]; the results indicated that restful sleep was associated with the absence of depressive symptoms and higher a GPA among college students. These authors recommended considering ways to show students the benefits of restful sleep for mental health and academic performance. For detailed evidence of all the processes involved at the different levels that relate restful sleep to the cognitive properties of the brain, which is outside the scope of this work, readers can review the references mentioned above; we have done no more than present some of the approaches found in the literature. The relationships among all the processes that link sleep with our memory are still far from clear, as noted by [125], however, these authors propose some strategies for improving memory by optimizing sleep, which could be used in applied contexts. The National Sleep Foundation (NSF) has reported guidelines and recommendations that should be taken into consideration during the COVID-19 pandemic [128].

### 3.4. Mindfulness as a Protective Factor and Learning Promoter

The fourth protective factor is related to contemplative practices, treated as a separate topic. Mindfulness can be described as a non-judgmental process which uses complete attention to what is occurring in the present, as open-heartedly as possible [129]. According to psychological and neuroscientific findings, it has been suggested that mindfulness involves enhanced self-regulation, self-awareness, and regulation of the emotions [87]. Mindfulness, also called mindfulness-based stress reduction (MBSR) [130], has been included in several fields of science, i.e., a search in Web of Science which included the term “mindfulness” obtained 17,738 results (27 March 2021). Mindfulness studies have included the neural mechanisms involved in this contemplative practice [87,131] and in physical education [132]; the application of mindfulness to reduce stress or anxiety and mental health disorders, and to enhance positive emotions, has also been widely evaluated [133,134].

Interestingly, a recent work showed that mindfulness interventions in adolescents caused a decrease in anxiety, but also produced changes in the network properties of the amygdala, increasing its structural connectivity [135]. Neuro-imaging studies have also shown changes in meditators and their brain connectivity due to mindfulness meditation [136]. The authors evaluated whether mindfulness meditation induced long-lasting effects in topological features in brain networks. The results demonstrated changes in the amygdala networks and also in the right hippocampus, which display a higher degree in the theta band, related to memory processes. Moreover, these authors concluded that the constant practice of mindfulness was associated with a long-lasting change in the topology of definite areas of the brain, suggesting that this meditative practice might be able to induce brain plasticity [136]. Moreover, it appears that a long period of mindfulness practice is not required to induce structural plasticity in the brain. In a recent work, [137] reported that short-term (<30 days) practice of mindfulness meditation induced grey matter plasticity, suggesting changes in the ventral posterior cingulate cortex which is a significant hub associated with self-awareness, emotion, cognition, and ageing. This agrees with a previous report in which a meditation-naïve participant submitted to an eight-week programme of mindfulness practice; anatomical magnetic resonance demonstrated an increase in grey matter concentration in the left hippocampus and also in the posterior cingulate cortex, the temporo-parietal junction, and the cerebellum, all areas of the brain involved in memory and learning processes, emotion regulation, self-referential processing, and perspective taking [138]. Structural and functional changes in the brain (increase in cortical thickness in the left precuneus and decreased amplitude of low-frequency fluctuation) due to mindfulness practices in meditation-naïve subjects have also been reported [139]. Interestingly, the decreases were stronger in participants who also showed the greatest decreases in signs of depression.

The practice of mindfulness uses breathing, thoughts, and senses at the same time, helping to focus attention and awareness on our own bodies [140]. This technique works at an emotional level (triggering a series of biological processes). It is widely suggested for the treatment of pathologies such as stress and anxiety and is a useful tool for confronting the limitations that generate states of depression, anxiety, or stress [141,142,143]. As college students are very susceptible to high health-risk behaviours, a recent work [144] demonstrated that mindfulness-based intervention has valuable effects in both improving mindfulness and reducing many such behaviours. Similar results were reported by [145], who evaluated an online mindfulness-based mental health support programme for students, evaluating its effects on mindful experiencing, perceived stress, emotion regulation strategies, self-compassion, negative affect, and quality of life. Application of mindfulness programmes resulted in a reduction of perceived stress, a decreased frequency and intensity of negative affect experiences, and an increase in being mindful in life.

As indicated at the beginning of this work, there is a large body of evidence that the COVID-19 pandemic has generated a profound psychological impact due to the restrictions imposed. Various studies have shown the applicability of mindfulness to treat mental problems associated with the COVID-19 outbreak. [146] reported that their results showed that mindfulness was beneficial to emerging adults (mean age 20.8 years) during the pandemic. Another interesting work reported that during the COVID-19 pandemic, emerging adults suffering social loneliness were less disposed to engage in mindfulness, resulting in an increase in depressive symptoms [147]. A similar result was reported by [148], who showed that a high disposition to mindfulness enhanced well-being and helped in dealing with stress or anxiety caused by the pandemic. The increased social media exposure of students during the pandemic has been a cause of severe psychological distress, since this phenomenon is associated with rumination, as was shown in a study carried out in 439 college students at two universities in Wuhan, Hubei Province, China [149]. According to the authors, mindfulness meditation was shown to be a protective factor, buffering the adverse effects of social media exposure on psychological distress in students. Mindfulness meditation by teachers has also had positive effects during the pandemic. Italian female teachers (*n* = 66, age 51.5 ± 7.9 years) subjected to mindfulness showed significant improvements in affective empathy, emotional exhaustion, anxiety, and depression, demonstrating that contemplative practices enhanced psychological well-being and interoceptive awareness [150]. Much information is provided in the literature about the positive effects of mindfulness on our wellness, and therefore it might be expected that this emotional well-being would also be associated with an improvement in learning. However, learning and practicing mindfulness is a long process which takes time and space, and future studies are recommended to focus on issues of sustainability and group differences. Learning mindfulness may take time, depending on the engagement and motivation of each individual.

An important question under stress situations (pandemic, e-learning teaching, lockdown, etc.) is teacher burnout. Teachers have had to transform in-person lesson plans into something that can be taught online. Moreover, they have had to learn new apps quickly, and have been subjected to emotional exhaustion due to disruptive student behaviour, a situation that had already been reported before the pandemic started [151]. Previous works have indicated that managing troublesome behaviour of students in the classroom contributes significantly to stress and emotional exhaustion in teachers, as well as low self-efficacy, enthusiasm, and job satisfaction [152,153]. This situation has probably been exacerbated by online teaching, where managing students from a distance is even more complicated. In this context, [154] evaluated the benefits of trauma-informed training and mindfulness practice on educator attitudes and burnout. The Attitudes Related to Trauma-Informed Care (ARTIC) scale and the Maslach Burnout Inventory were used by the authors to compare trauma-informed attitudes and burnout levels in a sample of 112 educators. The results showed that a combination of trauma-informed training and mindfulness practice significantly decreased emotional exhaustion in teachers, improving their self-efficacy as well as their personal achievement.

## 4. Concluding Remarks and Guidelines

It is well known that incorporating educational neuroscience into educational processes has been questioned, because education and neuroscience have different points of view regarding learning processes. Associations between these two areas of science are necessary to understand the complex processes involved in learning. In recent years, neuroscientific studies have been carried out related to restful sleep, nutrition, physical activity, and the practice of mindfulness to evaluate their importance in stimulating memory and cognition. This is especially relevant to education because it is closely linked with the positive consequences for the main actors, i.e., teachers and students, and the educational community in general. Undoubtedly, it would be a mistake for educational communities, and for society as a whole, not to benefit from the findings of neuroscience on learning. It is helpful to share evidence that promotes the greater well-being of people, even more so when we face challenging situations that put our mental well-being and ability to learn at risk, such as the COVID-19 pandemic.

Factors that protect learning transcend people’s cultural characteristics and personal beliefs, since no defined physical exercise, consumption of particular foods, or special sleeping conditions are promoted. Recreational physical activity can be suggested according to the age and physical possibilities of each individual. It must be a relaxing, positive activity that promotes a good mood or psychological state. Foods should contain proteins, omega 3, DHA, raw glucose, minerals, and fibre, within individual economic and social possibilities.

For rest factors, a place is needed that allows restful sleep, with the hours of sleep required appropriate to the individual’s age [155]. As mentioned above, deep sleep is not a strict requirement to achieve the consolidation of learning, given that similar consolidation mechanisms can be achieved in some waking states, such as quiet, restful wakefulness.

In addition, it is interesting to highlight the existence of bridges between education and the findings of neuroscience, for example, in the experiential learning theory of David Kolb [156]. This pedagogical proposal encourages discovery and interest in learning through enriched, diverse sensory stimulation (use of several senses), capable of crossing the negative emotional barriers that inhibit learning, and of promoting memory consolidation through practice of what has been learned. These connections between neuroscience and education are interesting and, although it is necessary to have professionals who integrate education and neuroscience, it is no less true that important inputs can be collected for the transfer of knowledge in general education, especially in the current context of a pandemic. Findings such as mindfulness practices can lead to positive effects on individual stress problems, facilitating learning by reducing the cognitive blocks caused by stress, activating the potential for learning, and favouring quality of life.

Given the diversity of fields involved in education and learning processes, and thus the different points of view as to what is meant by “learning”, we must accept opposing opinions regarding the application of a neuroscientific approach in education, extrapolating results from research to the dynamic and complex world of education. The present work is oriented towards a more holistic vision of individuals, considering all the dimensions of their lives such as qualitative improvements in the environment and the promotion of pleasant physical activity. The latter has epigenetic consequences in people, regardless of whether they are aware of this fact. In addition, understanding neuroscientific findings in learning is far from being a reductionist approach, since it establishes a very important validation of complete emotional and psychological well-being as a requirement for good learning.

We believe that the elements mentioned in this work, rather than being an “accurate recipe”, are a general invitation to all the actors in education to consider these aspects. They can suggest these recommendations as a positive modification of how people live, which could have a positive impact on the ability to learn, especially in difficult times like the present. It is clear that the protective factors proposed in this paper have not yet been incorporated into neuroscience models. However, each of them has been incorporated into studies (more or less deep), and they have been shown to exert some kind of influence, at the physiological or molecular level in the brain, on the development of learning and cognition. All the proposed factors have been shown to potentiate neural plasticity or long-term memory [83,85,90,94,108,110,122,125,137,139,157].

In the following paragraphs, we propose some guidelines for readers. Although they are not developed in this manuscript, we believe that they can be useful for teachers from a psychological point of view to encourage good cognitive development in their students.

E-learning sessions, in the context of COVID-19, should allow students to actively experience what they have learned. The senses can be favoured by assigning the students tasks such as observing their surroundings, using elements present in their own homes, observing plants or other elements in their environment, interviewing their parents, studying pet behaviour, contrasting weights or textures, etc. Using more cognitive resources available in the students’ homes to show them what we want to teach strengthens their understanding and future recall. Later on, the teacher could encourage genuine curiosity in the students, requesting brief written or verbal assignments at a distance, either individually or in groups, where they can classify what they have explored. When they try to conceptualize this into general categories, the teacher can incorporate hypothesis formulation as a first step, and subsequently active, literature-based experimentation that validates previous learning. In addition to experimental learning, it is essential to consider the students’ neural plasticity and attention span capacity (less than 30 min) [158]. The contents need to be “chunked” into smaller time periods leading to small learning units, which finally, build to a complete learning unit. Moreover, each period must be started by an activity to stimulate interest and curiosity in the student (positive emotions) and maximize the sense of control over what is learnt. It is important to repeat topics over time to stimulate long-term retention, promoting the development of neural networks that are easily stimulated later. In addition, it is necessary to evaluate the final product (responses to a questionnaire for example), and also the previous processes carried out by the student in order to answer the questionnaire (abstract, concept mapping, etc.). Given that e-learning sessions in schools or universities last 50 to 90 min, the teacher must be able to capture the students’ attention, as attention levels can vary according to motivation, mood, the perceived relevance of the topics, etc.

Knowledge transfer is favoured, on the one hand, by the activation of knowledge through questions, exploring interests, and summaries of previous content. These actions make it possible to capture attention and overcome emotional filters that block learning. On the other hand, the activation of pre-existing neural networks is facilitated by assembling them into new synapses, allowing the consolidation of new learning in the short-term memory.

The flipped classroom format is highly recommended as it actively engages students in their learning. It allows them to explore content prior to class, assembling new content based on that provided by the teacher [159]. A flipped classroom allows links to be created between existing neural networks and the acquisition of new content, by executing new synapses, and thus enhancing higher cognitive learning.

Incorporating the arts into the classroom (as part of study areas) favours the incorporation of executive functions in learning. In the context of COVID-19, therefore, it is recommended that teachers assign tasks that involve creativity (poetry, songs, videos, and documentary analysis) using the contents of the subjects to be learnt. This is important, because it has been reported that depression, anxiety, and stress, all associated with the pandemic, are strongly related with disruption of the executive functions [160].

Last but not least, it would be desirable for the whole educational community to put into practice at least some minimum element(s) of those suggested here, gradually and realistically adding brief goals. This would help to challenge demotivation in the high-stress context of the pandemic [13]. In such a context, it is normal for people to be divided between demotivation and the need to perform a task. Accomplishing small tasks helps them to overcome the emotional barriers that block learning, thus regaining trust and optimism. Therefore, we invite readers to promote a healthy life which includes new experiences and learning, despite the current circumstances. It is in our hands to do everything possible to build a better quality of life, introducing improvements into the transfer of knowledge in educational communities. The learning environment and the learning process should not be designed to allow students to learn in the same way and at the same level as in the classroom, but rather should be designed with diverse strategies in mind that favour the use of the greatest possible number of perceptual channels.

## Figures and Tables

**Figure 1 brainsci-11-00552-f001:**
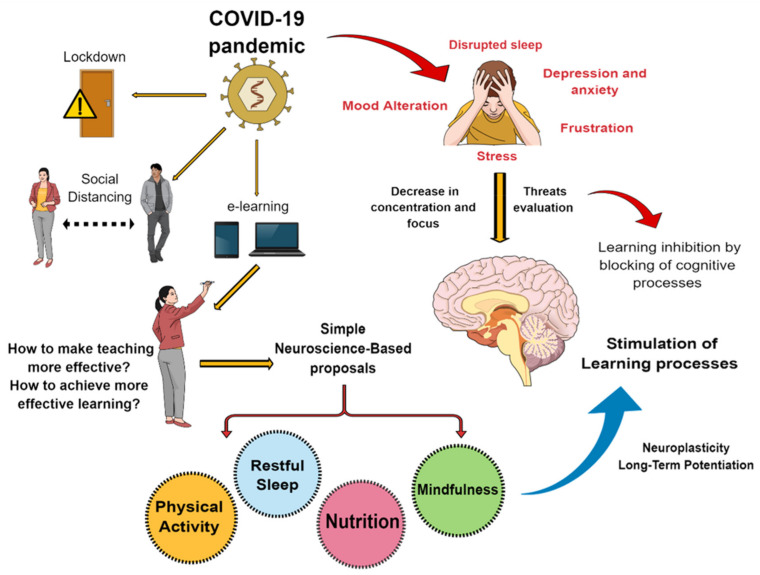
Potential psychological alterations due to the COVID-19 pandemic, effects on the cognitive processes, and simple neuroscience-based proposals used in brain resilience to stimulate learning processes.

**Figure 2 brainsci-11-00552-f002:**
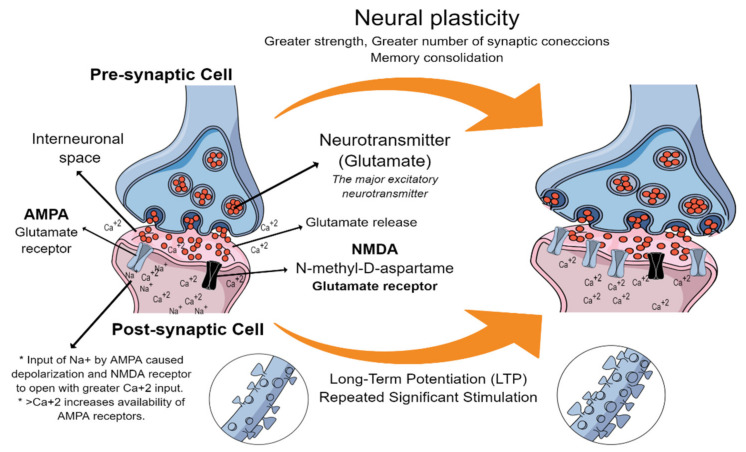
Simple representation of neural plasticity triggered by repeated significant stimulation (long-term potentiation).

**Figure 3 brainsci-11-00552-f003:**
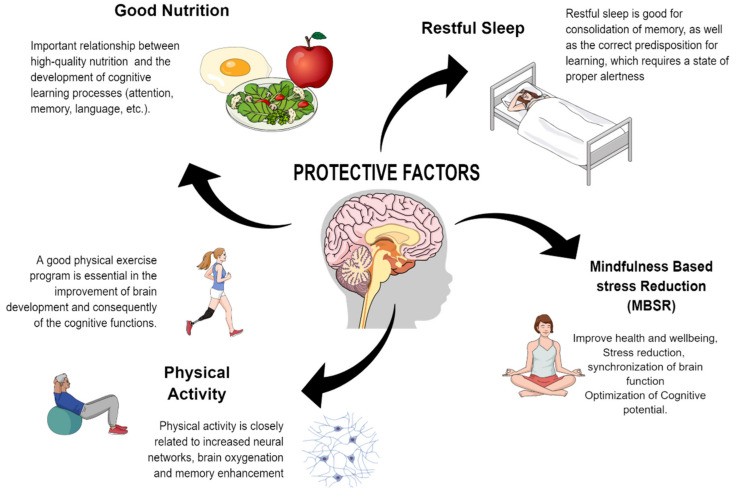
Proposed neuroscience-based protective factors that favour the stimulation of cognitive functions in the brain.

## Data Availability

Data sharing not applicable.

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
