# Peer review of "Mindfulness and Other Simple Neuroscience-Based Proposals to Promote the Learning Performance and Mental Health of Students during the COVID-19 Pandemic"

_brainsci, 2021, doi:10.3390/brainsci11050552_

Round 1

Reviewer 1 Report

The article presents an interesting and timely perspective about protective factors and cognitive enhancement practices to support study and learning in school contexts, of particular relevance during the COVID-19 pandemic. The protective and cognitive enhancement practices include physical activity, good sleep, nutrition and mindfulness, the latter being particularly highlighted.

The article appears potentially relevant and timely, also for its potential applications in society, with particular reference to this period with the COVID-19 pandemic, though with particular implications for the field of education. Howerver, as it will be explained more below, the neuroscientific and cognitive parts of the manuscript appear to lack depth. 

The section about the COVID-19 outbreak and adverse effects on learning appears supported by evidence and comprehensive, although there are some repetitions about COVID-19 related adverse effects and difficulties throughout other sections.

Also the section about the protective effects of mindfulness training appears well grounded on research findings, with a good review and addressing of relevant implications. 

One of the main problems of the article is that the neuroscientific bases are often generical/superficial, with no adequate references to and discussion of involved brain networks, core regions and processes, and do not reach the standards of Brain Sciences.  The suggested synergy between the four protective factors/practices is not addressed at the level of core processes and mechanisms, to be clarified in the main text and possibly in a dedicated figure. Neuroplasticity, long-term memory, memory consolidation, among other key cognitive and neuroscientific notions in the article, are not defined/characterized with precision and depth. As another instance, among others, the notion of learning is used ambiguously throughout the manuscript, e.g. with reference to school contexts (study) and more generically as related to basic cognitive, behavioral and neuroscientific processes/functions. Also the generic use of the term "neuroscience" could be better differentiated in terms of cognitive and affective neurosciences, or in terms of more fundamental neurobiology, depending on the (many) focuses through the manuscript. 

Another main problem is the organization of the text, with several repetitions and interpolations related to COVID-19 difficulties, long or non-sufficiently articulated paragraphs, lack of an overview about the article at the end of the Introduction, elements of the final guidelines which are not addressed/discussed in the sections before and/or related to neuroscience. There is a section dedicated to mindfulness, but the other three protective factors/practices are intermingled/mixed in the same section, without dedicated subsections and focuses. Also increased stress management, emotion regulation, and cognitive enhancement effects appear to some extent intermingled/entangled in several sections, and could possibly be better focused in subsections, together with the different protective factors/practices. 

Finally, the English needs to be revised/refined to some extent throughout all the sections of the article, including the title, in which "proposal" should probably be replaced by "proposals", or by a less generical terminology. 

Also with some checks of typos (e.g "sleep" rather than "slepp" in Figure 1). 

Author Response

COMMENT. The article presents an interesting and timely perspective about protective factors and cognitive enhancement practices to support study and learning in school contexts, of particular relevance during the COVID-19 pandemic. The protective and cognitive enhancement practices include physical activity, good sleep, nutrition and mindfulness, the latter being particularly highlighted. The article appears potentially relevant and timely, also for its potential applications in society, with particular reference to this period with the COVID-19 pandemic, though with particular implications for the field of education. However, as it will be explained more below, the neuroscientific and cognitive parts of the manuscript appear to lack depth.

ANSWER. Thank you very much for your comment. Several parts of the manuscript were complemented with additional information to increase the depth in the information. Fig. 2 was added. See Line 303-322; 333-336; 355-365; 409-422; 454-463; 501-525

COMMENT. The section about the COVID-19 outbreak and adverse effects on learning appears supported by evidence and comprehensive, although there are some repetitions about COVID-19 related adverse effects and difficulties throughout other sections.

ANSWER.  Thank you for the comment. The manuscript was revised and corrected to remove the excessive repetitions. See point 2.

COMMENT. Also, the section about the protective effects of mindfulness training appears well grounded on research findings, with a good review and addressing of relevant implications.

ANSWER.   Thank you for your comments. These sections were also enhanced, increasing their depth in the information. See lines 501-525

COMMENT. One of the main problems of the article is that the neuroscientific bases are often generical/superficial, with no adequate references to and discussion of involved brain networks, core regions and processes, and do not reach the standards of Brain Sciences.  The suggested synergy between the four protective factors/practices is not addressed at the level of core processes and mechanisms, to be clarified in the main text and possibly in a dedicated figure. Neuroplasticity, long-term memory, memory consolidation, among other key cognitive and neuroscientific notions in the article, are not defined/characterized with precision and depth. As another instance, among others, the notion of learning is used ambiguously throughout the manuscript, e.g., with reference to school contexts (study) and more generically as related to basic cognitive, behavioral and neuroscientific processes/functions. Also, the generic use of the term "neuroscience" could be better differentiated in terms of cognitive and affective neurosciences, or in terms of more fundamental neurobiology, depending on the (many) focuses through the manuscript.

ANSWER.  Thank you for your comments. We are in agreement with the comments. In the first instance, the manuscript was written in a simple way for a major understanding of no specialist readers. However, we addressed these comments, and it was added more information with neuroscientific bases. The significates of several processes (neuroplasticity, long-term memory, affective or cognitive neuroscience, among others were incorporated) see lines 96-101; 247 to 255; 274 to 280; It was also clarified that the term " learning " was used in the manuscript related in an educational context (lines 134-135). A figure was also added (New Fig. 2) to better outline the neuroplasticity process. See Lines 96-101; 303-322; 333-336; 355-365; 409-422; 454-463; 501-525.

COMMENT. Another main problem is the organization of the text, with several repetitions and interpolations related to COVID-19 difficulties, long or non-sufficiently articulated paragraphs, lack of an overview about the article at the end of the Introduction, elements of the final guidelines which are not addressed/discussed in the sections before and/or related to neuroscience. There is a section dedicated to mindfulness, but the other three protective factors/practices are intermingled/mixed in the same section, without dedicated subsections and focuses. Also increased stress management, emotion regulation, and cognitive enhancement effects appear to some extent intermingled/entangled in several sections, and could possibly be better focused in subsections, together with the different protective factors/practices.

ANSWER. Thank you for your comments. The manuscript text was revised and the repetitions were removed. An overview was added at the end of the introduction (see lines 102 to 119). Elements in final guidelines that were not addressed in the main text, were maintained, because we think that these can be a special contribution to the manuscript. However, this was stated in the manuscript, and reinforcing the idea that this paragraph has a more phycological point of view (see lines 641 to 644). Each protective factor was addressed in a subsection, as recommended.  See new sections 3.1, 3.2, 3.3 and 3.4.

COMMENT. Finally, the English needs to be revised/refined to some extent throughout all the sections of the article, including the title, in which "proposal" should probably be replaced by "proposals", or by a less generical terminology.

Also, with some checks of typos (e.g "sleep" rather than "slepp" in Figure 1).

ANSWER. The English was corrected by native speaker translator. The figure was corrected.

Reviewer 2 Report

Interesting paper that needs some improvements. 

The title should include better focused addressed: the paper sounds like a narrative review and could be good to report it in the title. 

The introduction could be improved including some other references as well:

  • https://doi.org/10.6092/2282-1619/mjcp-2533
  • https://doi.org/10.6092/2282-1619/mjcp-2424
  • https://doi.org/10.5114/hpr.2020.101249

Author Response

REVIEWER 2

COMMENT. The title should include better focused addressed: the paper sounds like a narrative review and could be good to report it in the title.

ANSWER. Thank you for your comments, The title was enhanced

COMMENT. The introduction could be improved including some other references as well:

https://doi.org/10.6092/2282-1619/mjcp-2533

https://doi.org/10.6092/2282-1619/mjcp-2424

https://doi.org/10.5114/hpr.2020.101249

ANSWER. The introduction was revised, and the suggested references were incorporated see references 8 , 10 and 11 in the manuscript.

Reviewer 3 Report

The manuscript offers a broad perspective and anecdotal survey for several contributing factors that may underlie diminished learning by school students during the COVID 19 pandemic social restrictions. The perspective sketches relationships that may exist across mood alterations, disrupted sleep, stress, and depression, which undermine the preparedness and vigilance of students to learn in person or via on-line instruction. The authors propose that all of these factors can be related directly to findings in neuroscience (lines 190-193).

The manuscript presents a perspective on an important current topic. Although not bringing specific insights for a brain sciences audience, the review could serve as starting point for hypotheses to be tested in education research.

The narrative framework of the manuscript loosely weaves general information about good nutrition, restful sleep, physical activity and mindfulness-based stress reduction (MBSR) as protective factors that can ameliorate environmental threats undermining students’ coping mechanisms, as above. Only one of the observations proposed as a protective factor, MBSR, has a direct implication correlated with studies of brain function. As such, the perspective offered by the authors speaks to a broader view of healthy lifestyle choices. Healthy lifestyle choices are implicated with cognitive function in many general population studies, but the present text presents no tight linkage between student learning data during the COVID 19 pandemic and controlled data evidencing cognitive declines.

The single data point referred to under MBSR as a protective factor draws on conclusions from a poster presentation of un-monitored on-line surveys (ref 107). Although there may be vetted results supporting predictive relationships between students’ cognitive performance and assessments of disruptions to their learning experiences during the COVID 19 pandemic, no results and analysis of this kind is included to support of the manuscript’s conclusions and recommendation.

Lesser concerns are that English usage, spelling and prose all need improvement. For examples, the text in lines 43 through 129 repeat many of the same ideas using many digressions. The text is not concise. See “Good Slepp” in Figure 1. Figures 1 and 2 do not relate to any data or specific neuroscience-based model of cognitive performance.

Author Response

REVIEWER 3

 COMMENT. The manuscript offers a broad perspective and anecdotal survey for several contributing factors that may underlie diminished learning by school students during the COVID 19 pandemic social restrictions. The perspective sketches relationships that may exist across mood alterations, disrupted sleep, stress, and depression, which undermine the preparedness and vigilance of students to learn in person or via on-line instruction. The authors propose that all of these factors can be related directly to findings in neuroscience (lines 190-193).

The manuscript presents a perspective on an important current topic. Although not bringing specific insights for a brain sciences audience, the review could serve as starting point for hypotheses to be tested in education research.

ANSWER. Thank you for your comments. We are in agreement with the idea that could serve as starting point for hypotheses to be tested in education research (see abstract line 26). This was stated in the text also for the knowledge of the readers. On the other hand, this topic is very broad, and has many points of view, and still many things to discover and study. However, everything related with protective factors, and its relationship with the neuroscience (physiological and molecular processes), was supported by scientific literature. In this same sense, for a major depth in the information, we added in each topic several sentences, which clarifies in a better way the relationship of these protective factors with neuroscience findings.  See lines 355-365; 409-422; 454-463; 501-525

COMMENT. The narrative framework of the manuscript loosely weaves general information about good nutrition, restful sleep, physical activity and mindfulness-based stress reduction (MBSR) as protective factors that can ameliorate environmental threats undermining students’ coping mechanisms, as above. Only one of the observations proposed as a protective factor, MBSR, has a direct implication correlated with studies of brain function. As such, the perspective offered by the authors speaks to a broader view of healthy lifestyle choices. Healthy lifestyle choices are implicated with cognitive function in many general population studies, but the present text presents no tight linkage between student learning data during the COVID 19 pandemic and controlled data evidencing cognitive declines.

The single data point referred to under MBSR as a protective factor draws on conclusions from a poster presentation of un-monitored on-line surveys (ref 107). Although there may be vetted results supporting predictive relationships between students’ cognitive performance and assessments of disruptions to their learning experiences during the COVID 19 pandemic, no results and analysis of this kind is included to support of the manuscript’s conclusions and recommendation.

ANSWER. Thank you for your comments. As it was mentioned above, the relation between these protective factors and the neuroscience, were supported for scientific studies that demonstrated some relation with the brain functions, specifically with the memory and cognition. In this sense, it was added more information to enhance the manuscript. On the other hand, we are in agreement with the studies have been carried out in general population studies. However, in the literature there is a close relationship between stress and the current pandemic, and the reduction in learning as a result of stress. This has been highlighted in the manuscript.  Since the pandemic is recent, the literature is still scarce. For this reason, we went back to do a search engine review, and we included new studies that were published after this manuscript was sent to the journal (references 43, 44, 50 and 51).

COMMENT. Lesser concerns are that English usage, spelling and prose all need improvement. For examples, the text in lines 43 through 129 repeat many of the same ideas using many digressions. The text is not concise. See “Good Slepp” in Figure 1. Figures 1 and 2 do not relate to any data or specific neuroscience-based model of cognitive performance.

ANSWER. Thank you for your comments. The language was enhanced and corrected by native speaker. The Good Slepp” was corrected. The figures were only used to shown in a simple way the relation between the effects of covid-19 pandemic and potential effects on learning processes in student (figure 1). The Figure 2 (now figure 3 in the new text), shows the four protective factors, and as previously mentioned, the relation between these protective factors and the neuroscience, supported for scientific studies that demonstrated some relation with the brain functions, specifically with the memory and cognition. In this sense, it was mentioned in the text the relation between four factors and the neuroplasticity (supported by scientific findings).

Round 2

Reviewer 1 Report

The article appears more clarified and better organized. 

Also its scope and objectives have been clarified, with improvements in the text. 

The article thus seems to provide an useful timely contribution in the field of education research, in particular with the current pandemic.